# Modulation Effects of *Sargassum pallidum* Extract on Hyperglycemia and Hyperlipidemia in Type 2 Diabetic Mice

**DOI:** 10.3390/foods12244409

**Published:** 2023-12-07

**Authors:** Xing Xie, Chun Chen, Xiong Fu

**Affiliations:** 1SCUT-Zhuhai Institute of Modern Industrial Innovation, School of Food Science and Engineering, South China University of Technology, Guangzhou 510640, Chinalfxfu@scut.edu.cn (X.F.); 2College of Health, Jiangxi Normal University, Nanchang 330022, China; 3School of Food Science and Engineering, Guangdong Province Key Laboratory for Green Processing of Natural Products and Product Safety, Engineering Research Center of Starch and Vegetable Protein Processing Ministry of Education, South China University of Technology, Guangzhou 510640, China

**Keywords:** *Sargassum pallidum*, gut microbiota, serum metabolites, lipid metabolism, glucose metabolism

## Abstract

The aim of this study was to investigate the antidiabetic effect of the extract from *Sargassum pallidum* (SPPE) on type 2 diabetes mellitus (T2DM) mice. SPPE treatment alleviated hyperglycemia, insulin resistance (IR), liver and pancreatic tissue damage, hyperlipidemia and hepatic oxidative stress resulting from T2DM. SPPE reversed phosphoenolpyruvate carboxylase (PEPCK) and hexokinase (HK) activities to improve gluconeogenesis and glycogen storage in the liver. Furthermore, SPPE modulated glucose metabolism by regulating the levels of mRNA expression involving the PI3K/Akt/FOXO1/G6pase/GLUT2 pathway and could inhibit fatty acid synthesis by reducing the gene expression levels of fatty acid synthase (FAS) and acetyl-CoA carboxylase-1 (ACC-1). A 16 sRNA analysis indicated that SPPE treatment also reversed gut dysbiosis by increasing the abundance of beneficial bacteria (*Bacteroides* and *Lactobacillus*) and suppressing the proliferation of harmful bacteria (*Enterococcus* and *Helicobacter*). Untargeted metabolomics results indicated that histidine metabolism, nicotinate and nicotinamide metabolism and fatty acid biosynthesis were significantly influenced by SPPE. Thus, SPPE may be applied as an effective dietary supplement or drug in the management of T2DM.

## 1. Introduction

Type 2 diabetes mellitus (T2DM) is a chronic disease characterized by hyperglycemia and insulin resistance, and it accounts for over 90% of diabetic cases [1]. As reported by the International Diabetes Federation, the number of diabetic patients was about 537 million in 2021, and underdeveloped and developing countries are experiencing a fast growth rate [2]. Diabetes patients with high blood glucose levels for a long time have an increased risk of diabetes complications, such as kidney disease, retinopathy, cardiopathy and so on [3]. Simultaneously, a large number of diabetes cases show high-blood-lipid profiles, which is the key factor resulting in cardiovascular diseases [4]. Moreover, antidiabetic drugs like insulin, acarbose and metformin exhibit serious side effects, leading to abdominal discomfort, diarrhea and weight gain [5]. Therefore, it is necessary to explore novel antidiabetic medicines that are safe and have high efficiency, and phenolics from natural plants have attracted much attention in this regard. 

The IRS1/PI3K/Akt pathway is the critical insulin signaling pathway to maintain glucose homeostasis [6]. Hyperglycemia and hypolipidemia will cause the excessive production of reactive oxygen species and oxidative stress damage [7]. In addition, accumulating studies have confirmed that the gut microbiota is closely related to the development of T2DM and may contribute to the modulation of glucose metabolism pathways [8]. In short, these are important targets for managing T2DM. Phenolics have been reported to exhibit excellent antidiabetic effects through various targets [9]. Kang et al. found that dieckol from *Ecklonia cava* significantly reduced blood glucose and serum insulin levels, increased the activities of antioxidant enzymes, and enhanced the Akt and AMPK phosphorylation of muscle in db/db mice [10]. The phenolic-rich extract from *Hypericum attenuatum* Choisy alleviated T2DM by modulating the AMPK/PI3K/Akt/GSK3β signaling pathway and regulating GLUT4 and PPARγ expression [11]. The phenolic extract of noni fruit could regulate glycolipid metabolism by altering the abundance of gut microbiota in T2DM rats [12]. 

*Sargassum pallidum*, which belongs to the genus *Sargassum*, is an edible brown seaweed distributed in Japan and coastal areas of China and has a long history of application as medicines to treat goiter, scrofula and edema [13]. *Sargassum pallidum* is abundant in phenolics, fatty acids, polysaccharides and terpenoids and possesses antioxidant, anti-inflammatory, hypoglycemic and anticancer activities [14]. Cao et al. observed that polysaccharides from *Sargassum pallidum* increased glucose consumption and attenuated insulin resistance by up-regulating PI3K and IRS-1 expression in HepG2 cells [13]. Our previous study also indicated that the phenolic extract of *Sargassum pallidum* displayed strong α-glucosidase inhibition [15]. Nevertheless, to date, the antidiabetic mechanism of the extract from *Sargassum pallidum* is not clear and deserves further investigation.

In this study, high-fat-diet- and streptozocin (HFD/STZ)-induced T2DM mice were used as a model. The effects of the phenolic extract from *Sargassum pallidum* on biochemical indices, histopathology, gut microbiota and serum metabolites in T2DM mice were evaluated. The gene expression involved in glycolipid metabolism in the liver was also investigated. This study presents a logical strategy to promote the application of *Sargassum pallidum* in the treatment of T2DM.

## 2. Materials and Methods

### 2.1. Chemicals and Materials

*Sargassum pallidum* powder was obtained from Qingdao City, Shandong Province. Streptozocin and metformin were purchased from Sigma Aldrich (St Louis, MO, USA). Triglyceride (TG), total cholesterol (TC), low- and high-density lipoprotein cholesterol detection (LDL-C and HDL-C), alanine aminotransferase (ALT), aspartate aminotransferase (AST), phosphoenolpyruvate carboxykinase (PEPCK), hexokinase (HK), catalase (CAT), glutathione peroxidase (GSH-Px), superoxide dismutase (SOD) and malondialdehyde (MDA) kits were provided by Nanjing Jiancheng Bioengineering Institute (Nanjing, China). 

### 2.2. Preparation of Extract from Sargassum Pallidum

Dried powder of *Sargassum pallidum* was mixed with 8.5% aqueous ethanol at a ratio of 1:30 (*m*:*v*) and extracted by ultrasonication (KQ-300DE, Kunshan ultrasonic instrument Co., Ltd., Kunshan, China) at 42 °C and 490 W for 55 min. The mixtures were filtered by centrifugation, and the residues were re-extracted once under the same conditions. All of the supernatants were collected, concentrated and lyophilized to obtain the extract of *Sargassum pallidum*. The total phenolic content of the extract was determined according to a previous method [15]. 

### 2.3. Animals and Experimental Design

Six-week-old male C57BL/6J mice were supplied by Guangdong Sijiajingda Biotechnology Co., Ltd. (Guangdong, China). All mice were acclimated for one week and given free access to water and chow at a specific temperature (22–24 °C) and humidity (50–60%) with a 12 h light/dark cycle. Mice in the control group (n = 8) were fed a low-sugar and low-fat diet, and those in the diabetic group (n = 50) were fed a 60% high-fat diet. After 4 weeks, all mice were fasted for 12 h, and the diabetic and control groups were, respectively, injected with STZ solution (120 mg/kg BW) or the same amount of normal saline. After 1 week, the fasting blood glucose (FBG) levels of mice were determined, and mice with values exceeding 11.1 mmol/L were regarded as T2DM. All T2DM mice were randomly divided into five groups (n = 8): model, SPPE50, SPPE150, SPPE250 and Met groups. The model group consisted of T2DM mice that were untreated except for normal saline. The experimental groups were orally administered 50, 150 or 250 mg/kg/day of SPPE, and the Met group was treated with 250 mg/kg/day of metformin; the treatments lasted for 4 weeks. The body weights and FBG levels of mice were recorded once a week. 

At the end of the experimental period, the feces of the mice were collected in cryogenic vials and quickly stored at −80 °C. After fasting overnight, the blood was obtained by abdominal aorta bleeding, and serum was obtained by centrifugation at 3500 r/min for 10 min. Then, mice were sacrificed; one part of the pancreas and liver was collected and kept at −80 °C, and another part was fixed with 4% paraformaldehyde for histopathological examination. All animal procedures were performed in accordance with the guidelines for the care and use of experimental animals in the laboratory animal center of Guangdong Pharmaceutical University and were approved by the Experimental Animal Ethics Committee. 

### 2.4. Oral Glucose Tolerance Test (OGTT)

In the last week, the mice were administered a 1 g/kg BW glucose solution by gavage after fasting for 12 h; then, the blood was collected from the tip of the tail, and the blood glucose levels were evaluated at 0, 30, 60, 90 and 120 min. The areas under the curve (AUCs) were calculated using Origin 2021 software. 

Determination of Serum Insulin Levels and HOMA-IR. On the last day, the blood samples were obtained after starvation overnight, and the fasting insulin levels were detected using an insulin ELISA kit (Nanjing Jiancheng Bioengineering Institute). Homeostasis model assessment insulin resistance (HOMA-IR) was calculated according to the formula reported by Wang et al. [16].

### 2.5. Biochemical Analysis

The concentrations of TG, TC, HDL-C, LDL-C, AST and ALT in the serum were measured using commercial assay kits. The liver tissue of the mice was mixed with suitable physiological saline and crushed into a homogenate, and then the supernatant was obtained by centrifugation. The TG and TC contents in the liver were also determined by the same methods as used for the serum. 

### 2.6. Histopathological Examination

The liver and pancreatic tissues were cut into 3–5 μm thick slices and stained with hematoxylin–eosin staining (H&E). Then, all images were observed and photographed with an orthostatic light microscope (NIKON ECLIPSE CI, Tokyo, Japan). 

### 2.7. Determination of Antioxidant Parameters and Key Glycometabolism Enzymes

The level of MDA and the activities of SOD, CAT and GSH in the supernatant of the liver tissue solution were determined by using commercial assay kits according to instructions. 

Briefly, 100 mg of liver tissue was mixed with the extracting solution at a ratio of 1:10 (*m*:*v*) and homogenized in an ice water bath for 5 min. After centrifugation for 15 min, the levels of PEPCK and HK in the supernatant were estimated by using commercial assay kits. The protein concentration of the samples was detected by using a BCA protein assay kit (Nanjing Jiancheng Bioengineering Institute). 

### 2.8. Quantitative Real-Time Polymerase Chain Reaction (RT-PCR) Assay

The total RNA of frozen liver tissues was extracted with Trizol reagent, and its concentration and purity were detected with a NanoDrop 1000 spectrophotometer (Wilmington, DE, USA). The cDNA was synthesized through reverse transcription and amplified by using an assay kit. The levels of cDNA were quantified using a Mini Opticon real-time PCR system (Bio-Rad, Hercules, CA, USA). The sequences of primers are described in Table 1. The relative expression levels of genes were evaluated by using the 2^−ΔΔCT^ method, and *β*-actin was selected as an internal standard. 

### 2.9. Cecal Microbiota Analysis

DNA extraction from mice feces was performed by using a QIAamp DNA stool mini kit (Qiagen, Germantown, MD, USA). The V3-V4 hypervariable regions of bacterial 16 rRNA genes were amplified with the following primers: 338F 5′-barcode-ACTCCTACGGGAGGCAGCAG-3′ and 806R 5′-GGACTACHVG GGTWTCTAAT-3′. The PCR products were sequenced with an Illumina MiSeq platform (Illumina, San Diego, CA, USA). The species information was collected by comparing the results with the Ribosomal Database Project. The data were further analyzed by using the online platform Personalbio Genes Cloud (www.genescloud.cn, accessed on 1 July 2023). After the quantification step, amplicons were pooled in equal amounts and sequenced at Shanghai Personal Biotechnology Co., Ltd. (Shanghai, China), using the Illumina MiSeq platform for paired-end 2 × 250 bp sequencing.

### 2.10. Untargeted Metabolome Analysis

The non-targeted metabolome analysis of serum was performed using LC-MS (Agilent, Santa Clara, CA, USA). The samples were separated with a HILIC column, and acetonitrile (A) and water containing 25 mM ammonium acetate and ammonia (B) were used as the mobile phase. The flow rate, injection volume and column temperature were 0.5 mL/min, 2 μL and 25 °C, respectively. The elution conditions were as follows: 0–0.5 min, 95% A; 7 min, 65% A; 8–9 min, 40% A; 9.1–12 min, 95% A. TOF MS data were scanned in the *m*/*z* 100–1000 in positive and negative ion modes. The other parameters for TOF MS were as follows: curtain gas, 30; source temperature, 600 °C; ion spray voltage floating, ±5500 V; declustering potential, ±60 V; Collision Energy, 35 ± 15 eV. The metabolites were identified by comparing retention times, molecular weights and mass spectrum fragments with the database.

### 2.11. Statistical Analysis

All data are shown as mean ± standard deviation (SD) and are from experiments repeated at least three times. One-way analysis of variance (ANOVA) and Tukey’s test were performed for statistical comparisons between groups, and a *p* value less than 0.05 was regarded as statistically significant (* *p* < 0.05, ** *p* < 0.01, *** *p* < 0.001). 

## 3. Results and Discussion

### 3.1. Phenolic Composition of SPPE

The total phenolic content of SPPE was 22.63 ± 0.21 mg GAE/g E. A previous study indicated that the main phenolics in *Sargassum pallidum* are 6-gingerol, quercetin-3-*O*-glucuronide, kuraridine and n-hexacosyl caffeatehexose, which are potential α-glucosidase inhibitors to treat T2DM [15]. In particular, quercetin-3-*O*-glucuronide had an α-glucosidase-inhibitory effect that was 20 times higher than that of acarbose [17]. 

### 3.2. Effect of SPPE on Body Weight

As shown in Figure 1A, the body weights of the model group exhibited a declining trend, while those of the SPPE150, SPPE250 and Met groups first decreased and then increased over the whole experimental period. And SPPE50 displayed a higher BW decrease than SPPER250. Compared with the model group, the body weights of the SPPE250 and Met groups were significantly enhanced after 4 weeks of treatment. The results indicated that a high dose of the phenolic extract from *Sargassum pallidum* could improve the weight loss of T2DM mice.

### 3.3. Effect of SPPE on Glucose Homeostasis

As shown in Figure 1B, the initial FBG levels in diabetic mice showed no significant differences among the groups. After supplementation with SPPE for 4 weeks, the FBG levels in the SPPE50, SPPE150, SPPE250 and Met groups were, respectively, decreased by 8.22%, 15.11%, 42.75% and 34.77%. A high dose of SPPE and Met treatment reduced the FBG level in T2DM mice. Therefore, SPPE250 was the only treatment that displayed significant effects on FBG compared to the model. For the insulin content, the insulin level in T2DM mice was higher than in normal mice, which suggested that diabetic mice had IR. Compared with the model group, the insulin levels in the SPPE and Met groups were reduced by 8.72–22.18% (*p* < 0.05), and that in the SPPE250 group was close to that in normal mice. As shown in Figure 1D, the HOMA-IR index of different groups ranged from 1.79 to 9.24 min mM (*p* < 0.05), and the values in the SPPE250 and Met groups were 2.43 and 2.03 times lower than that in the model group. The results indicated that SPPE could improve the FBG and insulin levels in T2DM mice, and the hypoglycemic effect of SPPE250 was similar to that of metformin. 

As given in Figure 1E, the blood glucose concentration in all groups reached the highest value at 30 min and then declined gradually, except for the model group, and the reduction in the SPPE250 group was more significant. The blood glucose levels in all T2DM mice at 120 min were higher than those at 0 min. For AUC profiles, the AUC values in the SPPE group were reduced in a dose-dependent manner. There were no marked differences (*p* > 0.05) between the SPPE50 and model groups. As shown in Figure 1F, the AUC levels in the SPPE250 and Met groups decreased by 860.25 and 727.13 min*mM compared with the model group. These findings suggested that SPPE treatment could increase glucose metabolism and insulin sensitivity in T2DM mice.

### 3.4. Effect of SPPE on Serum and Hepatic Lipid Profiles

To investigate the influence of SPPE treatment on hyperlipidemia, the levels of serum and hepatic lipids were evaluated. As displayed in Table 2, the serum TG, TC and LDL-c levels in the model group showed an increase compared with the control group and were decreased after SPPE treatment, while the trend of HDL-c levels was the opposite. In particular, for the SPPE250 group, the serum TG and LDL-c levels were 0.95 ± 0.06 and 0.52 ± 0.02 mmol/L, which were consistent with those in normal mice. In addition, compared with the model group, the concentrations of TG and TC in the liver also dropped, and the concentrations in the SPPE250 group were lower than those in the Met group. The results indicated that SPPE treatment had a lipid-lowering impact on T2DM mice.

### 3.5. Effect of SPPE on the Histomorphology of Liver and Pancreas Tissues

As shown in Figure 2A, the hepatocytes of normal mice were in an orderly arrangement with a clear structure, while those of the model group exhibited an irregular arrangement, fat accumulation and big vacuoles. With the increased intake of SPPE, the structure of liver tissue in T2DM mice was gradually clearer, and the number and size of fat vacuoles also decreased. Moreover, as shown in Figure 2B, the pancreas in the model group exhibited severe atrophy and degeneration with a small area, and the number of insulin cells was small. SPPE treatment repaired the injury to pancreatic tissue, and the recovery in the SPPE250 group was nearly to the level of the control group. The results indicated that the high-dose SPPE treatment could mitigate the liver and pancreas tissue damage caused by T2DM.

### 3.6. Effect of SPPE on AST, ALT and Oxidative Stress

As revealed in Figure 3A,B, in comparison with the model group, the AST and ALT levels in the SPPE groups were, respectively, reduced by 6.93–35.98% (*p* < 0.05) and 5.18–26.86% (*p* < 0.05) and showed a dose effect. The results indicated that SPPE could improve liver function in T2DM mice. 

As shown in Figure 3C–E, the SOD, CAT and GSH activities in the model group were weaker than those in other groups, and those in the SPPE250 group were enhanced by 16.06%, 63.90% and 58.90%, respectively. The MDA content in T2DM mice decreased in a dose-dependent manner after SPPE treatment. Specifically, the MDA level in the SPPE250 group was reduced by 26.33% compared with the model group. Moreover, the effect of SPPE and Met treatment at a dose of 250 mg/kg was not significant. The results suggested that SPPE treatment could strengthen the antioxidant enzyme capacity and alleviate oxidative stress in T2DM mice.

### 3.7. Effect of SPPE on Activities of PEPCK and HK

As illustrated in Figure 4A, the activity of PEPCK in the model group was stronger than in the control group, reflecting the occurrence of gluconeogenesis. After SPPE treatment, the activity of PEPCK in diabetic mice was reduced by 6.54–37.32%. Figure 4B shows that the activity of HK in the model group was weaker than in the other groups and increased by 13.23–65.70% after treatment with SPPE. The levels of PEPCK and HK in the SPPE250 and Met groups were high. These findings indicated that SPPE treatment could reverse phosphoenolpyruvate carboxylase (PEPCK) and hexokinase (HK) activities to improve gluconeogenesis and glycogen storage in the livers of T2DM mice. 

### 3.8. Effect of SPPE on mRNA Expression of Glucose and Lipid Metabolism Genes

As depicted in Figure 5A–E, compared with the control group, lower gene expression levels of PI3K, Akt and GLUT2 were detected in the model group, and the FOXO1 and G6pase expression levels were up-regulated. After 4 weeks of administration, SPPE treatment modulated the gene expression levels of those mRNAs involved in glucose metabolism. The gene expression of FOXO1 and G6pase in the SPPE250 group was reduced by 0.33 and 0.58 times, and that of PI3K, Akt and GLUT2 increased. Moreover, these mRNA expression levels showed no significant differences between the SPPE250 and Met groups, except for FOXO1 (*p* > 0.05). The results indicated that SPPE treatment may regulate glucose metabolism in the livers of T2DM mice by promoting glucose transport and preventing gluconeogenesis through the PI3K/Akt/FOXO1/G6pase/GLUT2 signaling pathway. 

As shown in Figure 5F,G, the mRNA expression levels of ACC-1 and FAS in the model group were higher than those in the control group. After treatment with SPPE, the gene expression levels of these two mRNAs declined in T2DM mice. Furthermore, the mRNA expression level of ACC-1 in the SPPE250 group was close to that in the control group. The results suggest that SPPE treatment could inhibit the expression of key lipid synthesis factors in T2DM mice. 

### 3.9. Effect of SPPE on Gut Microbiota

As given in Table 3, the Shannon and Chao indexes differed between the model and control groups, indicating that T2DM changed the composition of the gut microbiota in mice. In contrast, SPPE treatment showed insignificant effects on the richness and diversity of the microbiota in T2DM mice. A difference was found in diversity between the Met and model groups (*p* < 0.05).

The gut microbiota profiles of all groups at different taxonomic levels were further investigated. At the phylum level, *Firmicutes*, *Proteobacteria*, *Actinobacteria* and *Bacteroidetes* were the major bacterial communities in each group (Figure 6A). Compared with the model group, the level of *Proteobacteria* was reduced by 56.21%, and *Bacteroidetes* content was increased by 185.59%. The F/B value of 3.89 in the SPPE250 group was lower than those in the model and Met groups. At the genus level, compared with the model group, the lower contents of *Bacteroides*, *Lactobacillus* and *Parabacteroides* and higher contents of *Enterococcus* and *Helicobacter* were reversed after 250 mg/kg of SPPE treatment (Figure 6B). Furthermore, the relative contents of *Bacteroides* and *Lactobacillus* were augmented by 2.52 and 0.58 times in the SPPE250 group. The results suggest that SPPE treatment could regulate the abundance of beneficial and harmful bacteria in T2DM mice. 

### 3.10. Effect of SPPE on Serum Metabolites

The serum metabolites in four selected groups were analyzed by LC-MS. As shown in Figure 7A,B, the PCA scores indicate that the separation was good among the four groups in both negative and positive ion modes, and the quality of the data was acceptable for further study. Based on previous results, the SPPE250 treatment showed the best effect on diabetic mice in all groups, so this study focused on the SPPE250 and model groups. In comparison with the model group, the up- and down-regulation of metabolites in the SPPE250 group are visualized in volcano maps (Figure 7C,D), and the results suggest that 250 mg/kg of SPPE treatment had a significant effect on the metabolism of diabetic mice. 

After screening based on VIP scores > 1, the different metabolites in the two groups were displayed in a hierarchical clustering heat map (Figure 7E,F). A total of 17 and 12 metabolites were affected by SPPE treatment in the negative and positive ion modes, respectively. In the negative ion mode, compared with the model group, the levels of Succinate and Uric acid were increased, and the levels of Oleic acid, Heptadecanoic acid, Palmitic acid, 2-Hydroxy-3-methylbutyric acid and 11 other metabolites were significantly reduced. In the positive ion mode, the levels of seven metabolites were increased by SPPE treatment, mainly including Nicotinamide, Urocanic acid and Lincomycin. In addition, the levels of Thioetheramide-PC, MG (18:2(9Z,12Z)/0:0/0:0) [rac], Sphingomyelin (d18:1/18:0) and Dimethylglycine were decreased. Kyoto Encyclopedia of Genes and Genomes (KEGG) pathway analysis was performed based on the differential metabolites, and the enriched pathways in the model and SPPE250 groups are listed in Figure 7G. The results revealed that these metabolites are related to histidine metabolism, nicotinate and nicotinamide metabolism, and biosynthesis of unsaturated fatty acids and fatty acids, which may be important for SPPE to treat T2DM. 

### 3.11. Correlation Analysis between Gut Microbiota and Differential Metabolites

The complex relation between the gut microbiota and differential metabolites in the SPPE250 and model groups was visualized by Spearman’s correlation analyses (Figure 8). *Bacteroides* and *Lactobacillus* were positively correlated with Xanthine and 2-Methylbutyroylcarnitine and negatively correlated with Oleic acid, all cis-(6,9,12)-Linolenic acid and 1-Palmitoyl lysophosphatidic acid. Moreover, Heptadecanoic acid, Palmitic acid and MG (18:2(9Z,12Z)/0:0/0:0) [rac] were highly related to 11 bacterial communities (*p* < 0.05), including *Enterococcus*, *Oscillospira*, *Bifidobacterium*, *Ruminococcus*, *Odoribacter*, *Adlercreutzia*, *Parabacteroides* and so on. *Enterococcus* also exhibited a positive association with 2-Hydroxy-3-methylbutyric acid and Pantothenate. 

## 4. Discussion

T2DM is accompanied by IR, oxidative stress, an imbalance in the gut microbiota, and glucose and lipid metabolism disorders [18]. Synthetic drugs have toxic side effects when administered in the long term, while phenolics from natural plants have been proven as alternative drugs or supplements to prevent and treat T2DM. Our previous study found that SPPE showed excellent hypoglycemic activity in vitro, and 6-gingerol, quercetin-3-*O*-glucuronide, kuraridine and n-hexacosyl caffeatehexose were the main phenolics in *Sargassum pallidum* [15]. Therefore, HFD/STZ-induced T2DM mice were used as a model to further assess the hypoglycemic and hypolipidemic abilities of SPPE.

After SPPE treatment for 4 weeks, the weight loss of the model mice was reversed, and a high dose of SPPE was more significant. This result is similar to that reported by Liu et al., who found that Ginsenoside Rk3 treatment could increase the weight of T2DM mice [19]. Furthermore, the levels of FBG, insulin and HOMA-IR declined sharply in the SPPE250 and Met groups compared with the model group. The OGTT results suggested that the consumption of 250 mg/kg of SPPE enhanced the insulin sensitivity of T2DM mice. As previously reported, *Sargassum fusiforme* fucoidan and polyphenol-rich vinegar extract were effective in reducing FBG levels and improving glucose tolerance in T2DM mice, and the same phenomenon was observed in our study [20,21]. Among the phenolics from *Sargassum pallidum*, quercetin-3-O-glucuronide, 6-gingerol and poricoic acid A are the main components with strong inhibitory effects on α-glucosidase and may delay the digestion of carbohydrates, resulting in a decrease in glucose absorption in the small intestine. In addition, the pancreas is an important tissue in the maintenance of glucose homeostasis [22]. SPPE treatment increased the β-cell mass in the pancreas and prevented the degeneration of islets, which could contribute to improving pancreatic damage and FBG levels in T2DM mice. 

The development of T2DM can disturb lipid metabolism and result in dyslipidemia [23]. The serum lipid profiles showed that SPPE treatment could modulate dyslipidemia by significantly reducing the levels of TG, TC and LDL-c while increasing the level of HDL-c in T2DM mice. Wu et al. [24] confirmed that the ethanol extract of *Sargarsum fusiforme* normalized the lipid levels in T2DM mice, which is in line with our study. Quercetin derivatives, as the main phenolics in *Sargassum pallidum*, have been demonstrated to improve lipid metabolism by accelerating β-oxidation in mice [25]. Based on the combined data, SPPE showed strong hypoglycemic and hypolipidemic effects on T2DM mice.

As is well known, the liver is a vital organ involved in glycogen synthesis, gluconeogenesis, IR, lipid synthesis and oxidation [26]. Liver damage can reflect the glucose and lipid metabolism abilities of T2DM mice. AST and ALT are markers of liver function and are often used to evaluate it [27]. In this research, SPPE treatment decreased the serum AST and ALT contents in T2DM mice. Moreover, histomorphological observations of liver tissue showed that SPPE could reduce the number of fat vacuoles and fat accumulation. These findings suggested that SPPE could repair liver injury caused by T2DM in mice. In addition, excessive ROS generation in DM patients leads to oxidative stress and is highly related to IR and liver damage [28]. Our results showed that higher SOD, CAT and GSH levels and lower MDA content were detected in the SPPE group, and the effects in the SPPE250 and Met250 groups were comparable, indicating that SPPE may alleviate oxidative stress by enhancing antioxidant enzyme activities. As reported by Gheda et al., phenolic extracts from *Cystoseira compressa* also improved oxidative stress injury in the livers of diabetic rats by increasing MDA and GSH levels [29]. In addition, PEPCK and HK are important enzymes in gluconeogenesis, glycogen synthesis and glycolysis and are associated with glucose homeostasis [6]. SPPE could significantly decrease the PEPCK and HK concentrations to reduce gluconeogenesis and increase glycogen synthesis, acting to maintain hepatic glucose homeostasis and blood glucose levels. These results indicate that SPPE can regulate glycolipid metabolism in the livers of T2DM mice in various ways.

In order to explore the underlying mechanism, the expression of key mRNAs was evaluated to clarify the effect of SPPE treatment on hepatic glucose and lipid metabolism in T2DM mice. The PI3K-Akt pathway plays an important role in regulating glucose metabolism in T2DM; G6pase and FOXO are key enzymes for glucose synthesis and are implicated in hepatic gluconeogenesis [30,31]. GLUT2 is responsible for the transport of glucose and can alleviate IR in the body [32]. The consumption of SPPE up-regulated the expression levels of PI3K, Akt and GLUT2 and down-regulated the expression levels of FOXO1 and G6pase, implying that the PI3K/Akt/FOXO1/G6pase/GLUT2 pathway may contribute to the hypoglycemic effect of SPPE. A similar result was obtained using the phenolic extract from brown rice, which activated the PI3K/Akt signaling pathway to improve hepatic glucose metabolism [33]. In addition, FAS and ACC-1 are important enzymes involved in lipid metabolism, which, respectively, participate in catalyzing the synthesis of long-chain fatty acids into adipose tissue and fatty acid synthesis [34]. Lower expression levels of ACC-1 and FAS were observed in T2DM mice after treatment with SPPE, and these mRNA levels in the SPPE250 group were close to those in the normal group, indicating that SPPE could modulate liver lipid metabolism by suppressing fatty acid synthesis and fat accumulation. These results reveal that the antidiabetic mechanism of SPPE occurs through the modulation of the expression of key genes related to the PI3K/Akt pathway and lipogenesis in the liver. SPPE is rich in phenolics like 6-gingerol and quercetin derivatives. Some studies have reported that 6-gingerol and quercetin derivatives could enhance the mRNA expression of the PI3k/Akt pathway to alleviate insulin resistance in T2DM mice [35,36] and decrease lipogenesis by down-regulating the mRNA levels of ACC and FAS in mice [37,38]. Therefore, the restored expression of these glycolipid-metabolism-related mRNAs in T2DM mice may be due to the combined synergetic effect of the bioactive compounds, including phenolics, in *Sargassum pallidum*. 

A growing list of studies have shown that the composition and diversity of the gut microtia play critical roles in the development of T2DM [39]. *Proteobacteria* are abundant in DM patients and can result in an inflammatory response in the body [40]. *Bacteroidetes* and the F/B value were positively related to the plasma glucose concentration and lipid metabolism order [41]. In this study, the effect of SPPE on diversity and richness was not significant. Increased levels of *Bacteroidetes* and decreased levels of *Proteobacteria* and F/B were observed in T2DM mice after a high dose of SPPE treatment. Xia et al. [21] also found that a polyphenol-rich vinegar extract showed an antidiabetic effect by regulating the levels of *Proteobacteria* and F/B. *Lactobacillus*, as beneficial bacteria, can modulate glycolipid metabolism by mitigating intestinal inflammation to suppress metabolic diseases like DM [42]. *Parabacteroides* contributed to improvements in IR, inflammation and intestinal integrity [43]. *Enterococcus* and *Helicobacter* were highly associated with the occurrence of intestinal diseases [44,45]. After SPPE treatment, *Lactobacillus* and *Parabacteroides* were enriched, while the abundance of *Enterococcus* and *Helicobacter* declined in T2DM mice. Zhang et al. [46] reported that RBDF phenolics could reduce the level of *Enterococcu* in db/db diabetic mice. Qi et al. confirmed that Fu brick tea aqueous extract treatment increased the levels of *Lactobacillus* and *Parabacteroides,* exhibiting anti-inflammatory and hypoglycemic effects [47]. These results imply that SPPE could change the abundance of beneficial and harmful bacteria to exert an antidiabetic effect. 

Serum metabolites are the end products of gene expression involved in the regulation of metabolism and can reflect the effect of SPPE on the metabolism of T2DM mice. Niacinamide is the product of niacin amidation and is closely associated with glucose glycolysis and fat metabolism [48]. Supplementation with SPPE reversed the levels of Niacinamide and Succinate, thus regulating nicotinate and nicotinamide metabolism. Histidine metabolism could improve oxidative stress and inflammatory responses in the body [49]. After SPPE treatment, the levels of histidine metabolites like L-Anserine and Urocanic acid were increased, which may enhance the antioxidant enzyme activities and inhibit inflammatory reactions in T2DM mice. Sphingomyelin (d18:1/18:0), as a sphingolipid, is positively related to insulin resistance, β-cell dysfunction and inflammation [50]. The consumption of SPPE significantly reduced the level of Sphingomyelin (d18:1/18:0) in T2DM mice and may contribute to the balance of glucose homeostasis. Moreover, after SPPE treatment, lower levels of saturated fatty acids, like Palmitic and Heptadecanoic acid, were observed in T2DM mice, and the content of Oleic acid slightly declined. SPPE may inhibit the biosynthesis of saturated fatty acids and thereby reduce the risk of T2DM, with the same phenomenon also found by Liu et al. [51], but could not promote the production of unsaturated fatty acids like Oleic acid. Dimethylglycine is positively related to HOMA-IR and incident T2D [52], and SPPE treatment decreased the concentration of dimethylglycine in T2DM mice. In addition, Spearman’s correlation analyses showed that *Lactobacillus* was positively correlated with 2-Methylbutyroyl carnitine, suggesting that it may play a vital role in the improvement of fatty acid metabolism disorders [53]. *Enterococcus* was positively related to Heptadecanoic and Palmitic acids and may enhance the synthesis of saturated fatty acids. These findings indicate that SPPE could modulate serum metabolite disorders caused by T2DM, and the gut microbiota had a significant effect on the composition of serum metabolites. This suggests that SPPE may be used as a functional food for managing diabetes. However, which metabolites play the most important role still needs to be explored.

## 5. Conclusions

In conclusion, the present study indicated that SPPE could modulate glycolipid metabolism in T2DM mice. In addition, SPPE could improve gut dysbiosis by increasing the abundance of *Bacteroides*, *Lactobacillus* and *Parabacteroides* and reducing the abundance of *Enterococcus* and *Helicobacter*. The untargeted metabolome analysis indicated that a high dose of SPPE significantly changed the concentrations of 29 metabolites in diabetic mice and displayed a moderating effect on histidine metabolism, nicotinate and nicotinamide metabolism and fatty acid biosynthesis. The gut microbiota exhibited a high correlation with serum metabolites. These findings could provide a scientific basis for the clinical analysis of SPPE in T2DM.

## Figures and Tables

**Figure 1 foods-12-04409-f001:**
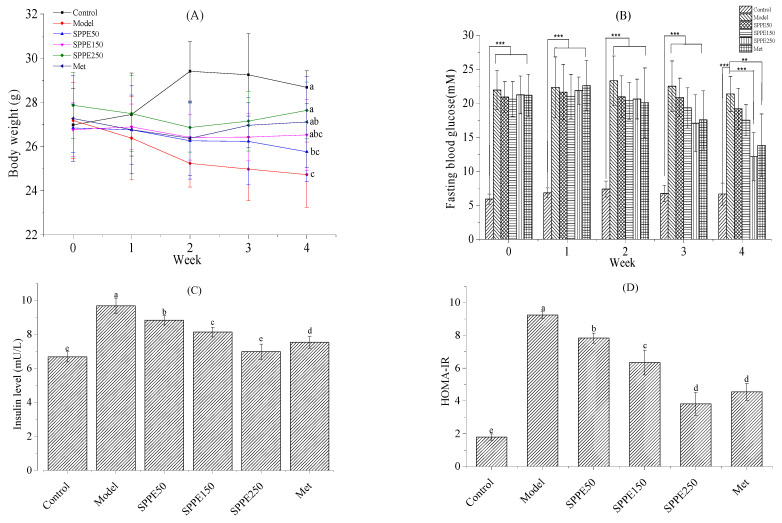
Effect of SPPE on the diabetes-related body indexes of T2DM mice. (**A**) Body weight, (**B**) fasting blood glucose, (**C**) insulin level, (**D**) HOMA-IR, (**E**) oral glucose tolerance test and (**F**) AUC. Different letters in the same pattern represent a significant difference (*p* < 0.05). ** and *** refers to the significance with *p* < 0.01 and *p* < 0.001, respectively.

**Figure 2 foods-12-04409-f002:**
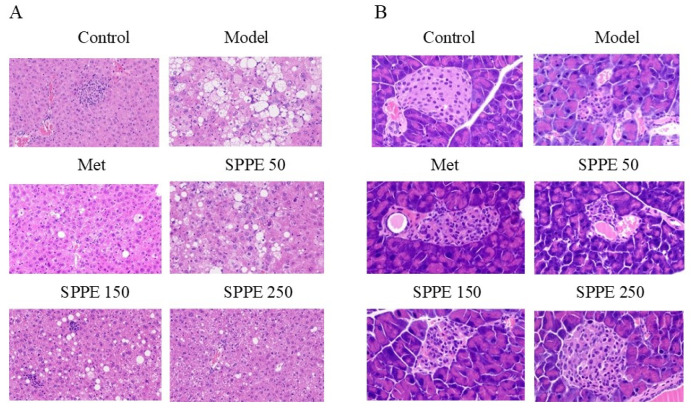
Effect of SPPE on the organs, including liver fat accumulation (**A**) and pancreatic tissue inflammation (**B**), in HFD/STZ-induced T2DM mice.

**Figure 3 foods-12-04409-f003:**
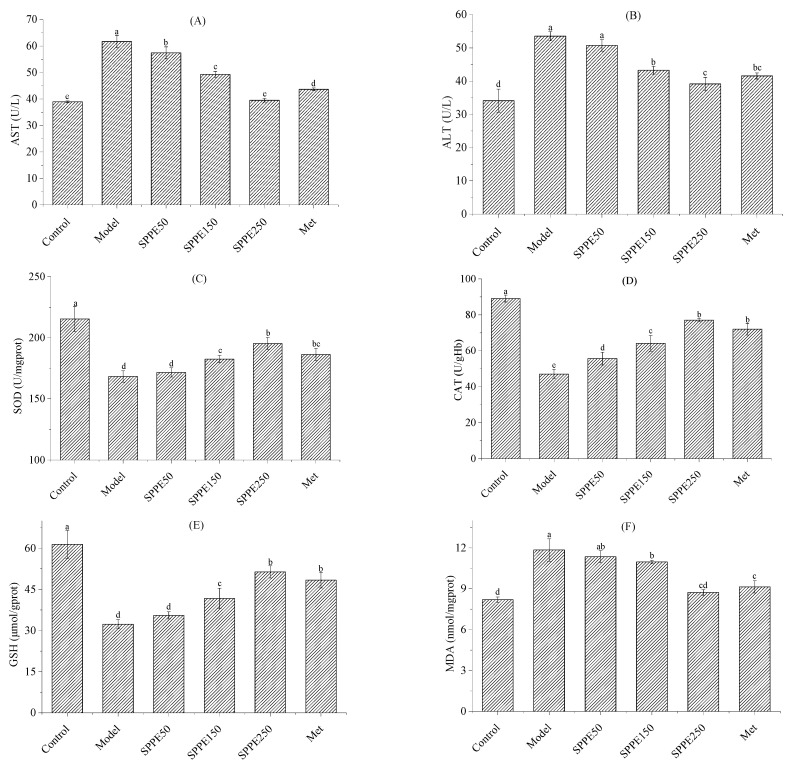
Effect of SPPE on serum AST (**A**), serum ALT (**B**), hepatic SOD (**C**), hepatic CAT (**D**), hepatic GSH (**E**) and hepatic MDA (**F**) levels in HFD/STZ-induced T2DM mice. Different letters in the same pattern represent significant differences (*p* < 0.05).

**Figure 4 foods-12-04409-f004:**
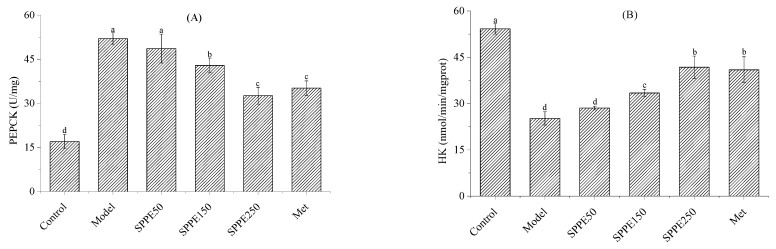
Effect of SPPE on the activity of hepatic PEPCK (**A**) and HK (**B**) in HFD/STZ-induced T2DM mice. Different letters in the same pattern represent significant differences (*p* < 0.05).

**Figure 5 foods-12-04409-f005:**
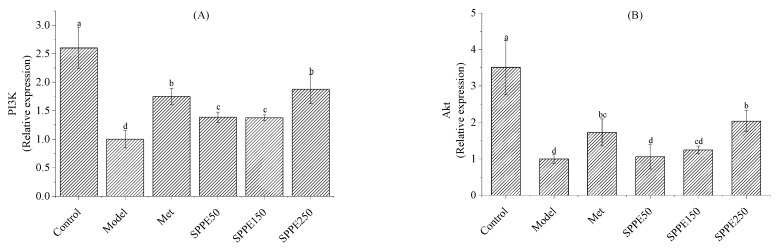
Effect of SPPE treatment on mRNA expression levels of PI3K (**A**), Akt (**B**), FOXO1 (**C**), G6pase (**D**), GLUT2 (**E**), FAS (**F**) and ACC-1 (**G**) in the livers of HFD/STZ-induced T2DM mice. Different letters are significantly different at *p* < 0.05.

**Figure 6 foods-12-04409-f006:**
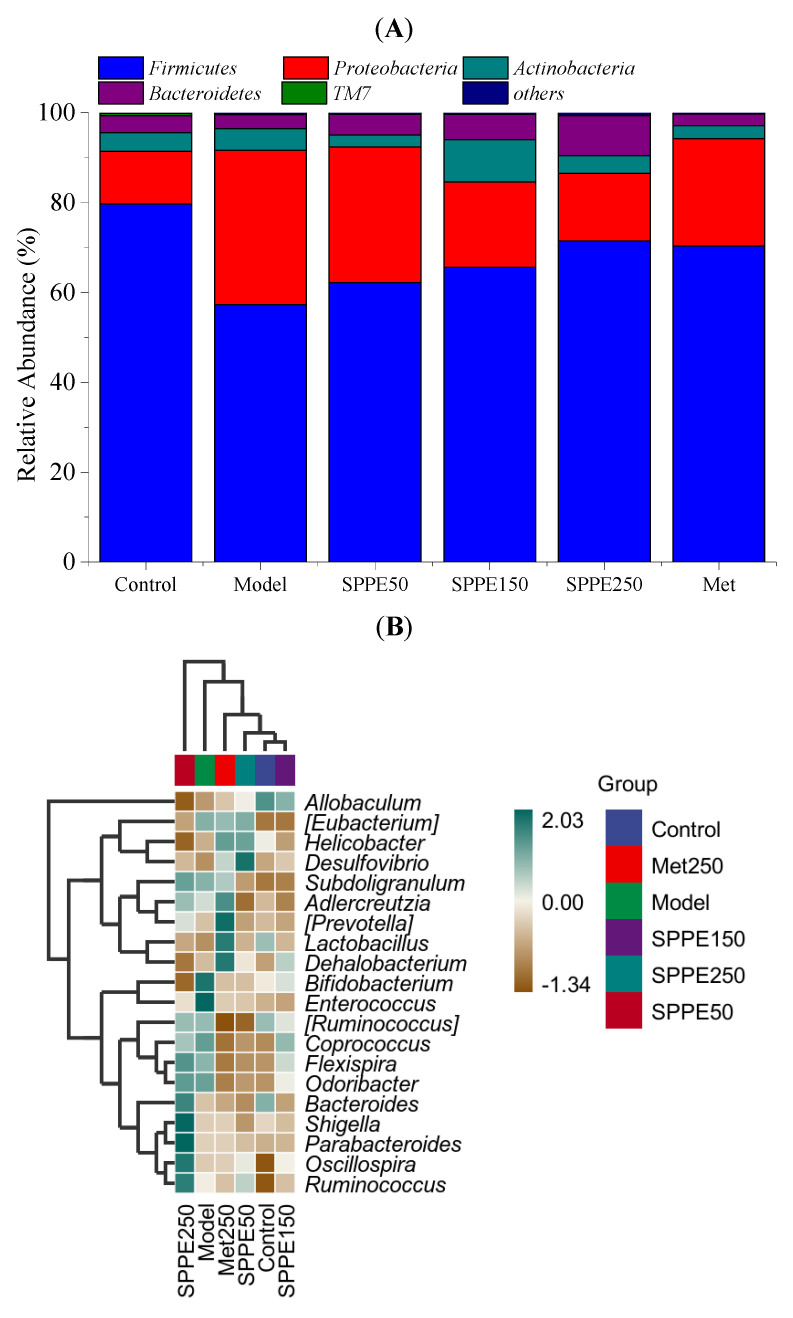
Effect of SPPE treatment on composition of gut microbiota in HFD/STZ-induced T2DM mice. At the phylum level (**A**); heat map of microbial composition at the genus level (**B**).

**Figure 7 foods-12-04409-f007:**
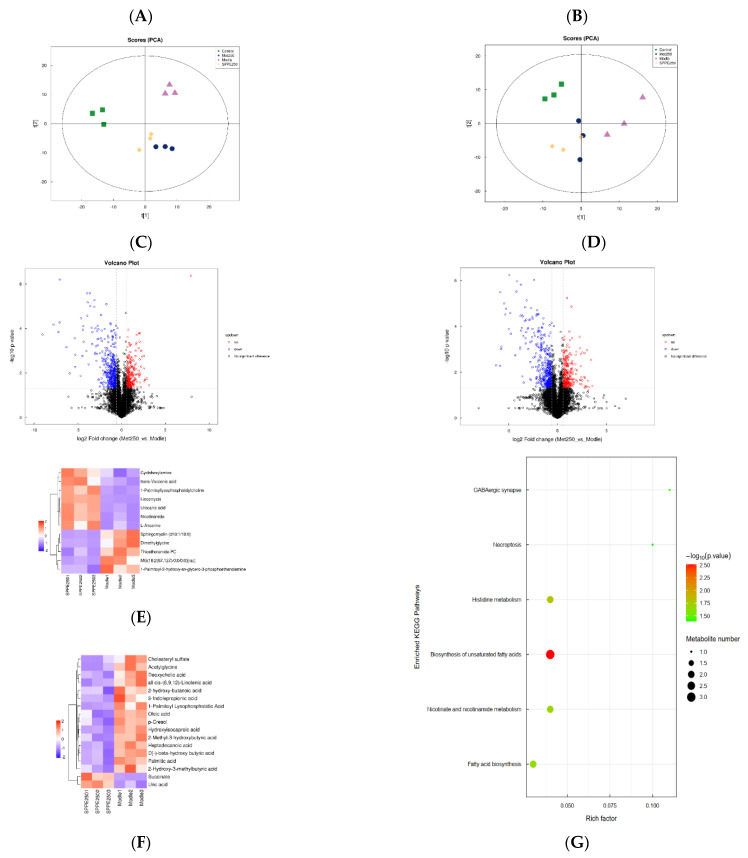
Effect of SPPE treatment on serum metabolites in HFD/STZ-induced T2DM mice. Scatter plot of PCA scores in the negative (**A**) and positive ion modes (**B**). Volcano plot for model vs. SPPE250 groups in the negative (**C**) and positive ion modes (**D**). Hierarchical clustering heat map of significant metabolites for model and SPPE250 groups in the negative (**E**) and positive ion modes (**F**). KEGG enrichment analysis for the model and SPPE250 groups (**G**).

**Figure 8 foods-12-04409-f008:**
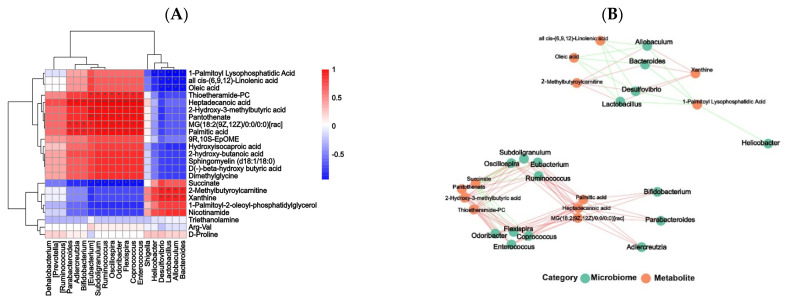
Correlation analysis between serum metabolites and gut microbiota at the genus level. Spearman’s analyses (**A**) and visualization of associated network (**B**) for model and SPPE250 groups. * *p* ≤ 0.05, ** *p* ≤ 0.01.

**Table 1 foods-12-04409-t001:** RT-PCR primer sequences.

Gene	Primer	Sequence (5′ → 3′)
PI3K	sense	ACACCACGGTTTGGACTATGG
	antisense	GGCTACAGT AGTGGGCTTGG
Akt	sense	GCCGGTGACAGACGATACT
	antisense	TGGCATTCACGTTTGTGGAGC
FOXO1	sense	GAGTTAGTGAGCAGGCTACAT
	antisense	TTTGGACTGCTCCTCAGTTCC
G6pase	sense	GGAGTCTTGTCAGGCATTGCT
	antisense	CGGAGGCTGGCATTGTAGAT
GLUT2	sense	GATCGCTCCAACCACACTCA
	antisense	CTGAGGCCAGCAA TC TGACTA
ACC-1	sense	CGCCAACAATGGTATTGCAGC
	antisense	TCG GATTGCACGTTCATTTCG
FAS	sense	GCGGGTTCGTGAAACTGATAA
	antisense	GCAAAATGGGCCTCCTTGATA
*β*-Actin	sense	GATCGATGCCGGTGCTAAGA
	antisense	TCCTATGGGAGAACGGCAGA

**Table 2 foods-12-04409-t002:** Effect of treatments on lipid profiles of serum TG, serum TC, serum HDL-c, serum LDL-c, hepatic TG and hepatic TC in HFD/STZ-induced T2DM mice.

Group	Serum TG (mmol/L)	Serum TC (mmol/L)	Serum HDL-c (mmol/L)	Serum LDL-c (mmol/L)	Hepatic TG (mmol/L)	Hepatic TC (mmol/L)
Control	0.98 ± 0.05 ^d^	2.85 ± 0.22 ^d^	2.33 ± 0.10 ^a^	0.50 ± 0.03 ^cd^	0.26 ± 0.01 ^e^	0.13 ± 0.01 ^e^
Model	3.36 ± 0.06 ^a^	6.37 ± 0.37 ^a^	0.82 ± 0.02 ^f^	0.78 ± 0.05 ^a^	0.50 ± 0.01 ^a^	0.40 ± 0.01 ^a^
SPPE50	2.71 ± 0.12 ^b^	5.68 ± 0.30 ^b^	1.15 ± 0.03 ^e^	0.70 ± 0.13 ^ab^	0.44 ± 0.03 ^b^	0.29 ± 0.03 ^b^
SPPE150	1.42 ± 0.15 ^c^	5.10 ± 0.01 ^b^	1.42 ± 0.02 ^d^	0.63 ± 0.08 ^bc^	0.38 ± 0.02 ^c^	0.22 ± 0.02 ^c^
SPPE250	0.95 ± 0.06 ^d^	4.06 ± 0.07 ^c^	1.98 ± 0.07 ^b^	0.47 ± 0.08 ^d^	0.26 ± 0.03 ^e^	0.16 ± 0.01 ^d^
Met	1.34 ± 0.11 ^c^	4.36 ± 0.19 ^c^	1.76 ± 0.14 ^c^	0.52 ± 0.02 ^cd^	0.32 ± 0.01 ^d^	0.18 ± 0.03 ^d^

Note: Different letters in the same column represent significant differences (*p* < 0.05).

**Table 3 foods-12-04409-t003:** Effect of SPPE treatment on alpha diversity of gut microbiota in HFD/STZ-induced T2DM mice.

Group	Index	
	Chao	Shannon
Control	767.08 ± 35.34 ^c^	5.20 ± 0.25 ^b^
Model	916.16 ± 65.12 ^b^	6.19 ± 0.23 ^a^
SPPE50	952.16 ± 38.82 ^b^	5.90 ± 0.15 ^a^
SPPE150	966.26 ± 48.21 ^b^	5.97 ± 0.21 ^a^
SPPE250	1004.20 ± 55.62 ^a^	6.72 ± 0.31 ^a^
Met	652.75 ± 45.32 ^d^	4.60 ± 0.18 ^c^

Note: Different letters in the same column represent significant differences (*p* < 0.05).

## Data Availability

Data are contained within the article.

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
