# Peer review of "Modulation Effects of Sargassum pallidum Extract on Hyperglycemia and Hyperlipidemia in Type 2 Diabetic Mice"

_foods, 2023, doi:10.3390/foods12244409_

Round 1

Reviewer 1 Report

Comments and Suggestions for Authors

Overall, your manuscript provides valuable insights into the potential therapeutic effects of Sargassum pallidum phenolic extract (SPPE) on type 2 diabetes mellitus (T2DM) in mice. Here are some comments and suggestions to improve the clarity and structure of your manuscript:

General Comments:

  1. Title and Abstract:
    • Ensure that the title accurately reflects the content of the manuscript.
    • The abstract is a crucial part of the manuscript. It should provide a concise summary of the background, methods, results, and conclusions.
  2. Organization:
    • Consider reorganizing the manuscript to follow the conventional structure of scientific papers: Introduction, Materials and Methods, Results, Discussion, Conclusion.

Specific Comments:

  1. Introduction:
    • Provide a more comprehensive introduction that clearly outlines the background, significance, and objectives of the study.
    • State the research hypothesis or questions you aimed to address.
  2. Methods:
    • Include details about the sample size, randomization, and blinding methods used in the study.
    • Clarify the methods for gene expression analysis, microbiota analysis, and metabolite analysis.
  3. Results:
    • Present the results in a more organized manner, possibly separating them into subsections for clarity.
    • Use tables and figures effectively to convey information without overwhelming the reader.
  4. Figures and Tables:
    • Ensure that figures and tables are appropriately labeled and referred to in the text.
    • Consider providing more detailed figure legends to aid the reader in understanding the content.
  5. Statistical Analysis:
    • Clearly state the statistical methods used, including significance levels.
    • Report p-values for all statistical tests performed.
  6. Discussion:
    • Relate your findings back to existing literature and discuss the implications of your results.
    • Address limitations and potential sources of bias in your study.
    • Consider including a subsection on the potential clinical applications of SPPE.
  7. Conclusion:
    • Provide a concise summary of your key findings.
    • Mention avenues for future research based on your results.

Comments on the Quality of English Language

no issues

Author Response

Dear reviewer,

We greatly appreciate the detailed constructive comments and suggestions from the reviewer. Those comments are very valuable and helpful for improving our paper. The manuscript has been carefully revised accordingly to the comments and suggestions are listed below.

  1. Title and Abstract

Ensure that the title accurately reflects the content of the manuscript. The abstract is a crucial part of the manuscript. It should provide a concise summary of the background, methods, results, and conclusions.

Response: The title has been revised to accurately reflects the content of the manuscript. The abstract is concise.

  1. Organization

Consider reorganizing the manuscript to follow the conventional structure of scientific papers: Introduction, Materials and Methods, Results, Discussion, Conclusion.

Response: The manuscript has been reorganized that the “Conclusion” part has been added.

  1. Introduction

Provide a more comprehensive introduction that clearly outlines the background, significance, and objectives of the study. State the research hypothesis or questions you aimed to address.

Response: The background, significance, and objectives of the study were stated in lines 51-53. The research hypothesis was stated in lines 76-78.

  1. Methods

Include details about the sample size, randomization, and blinding methods used in the study. Clarify the methods for gene expression analysis, microbiota analysis, and metabolite analysis.

Response: The gene expression analysis, microbiota analysis, and metabolite analysis have been stated and added in lines 162-164, 172-175, and 185-187.

  1. Results

Present the results in a more organized manner, possibly separating them into subsections for clarity. Use tables and figures effectively to convey information without overwhelming the reader.

Response: The results organization is based on their comprehensive analysis. The results are all presented using tables or figures.

  1. Figures and Tables

Ensure that figures and tables are appropriately labeled and referred to in the text.

Consider providing more detailed figure legends to aid the reader in understanding the content.

Response: More detailed information has been provided in Figure 1 and 2.

  1. Statistical Analysis

Clearly state the statistical methods used, including significance levels.

Report p-values for all statistical tests performed.

Response: The statistical analysis has been revised as comments in lines 189-192.

  1. Discussion

Relate your findings back to existing literature and discuss the implications of your results. Address limitations and potential sources of bias in your study. Consider including a subsection on the potential clinical applications of SPPE.

Response: In the discussion, the implications were stated. Also the findings, applications and limitations were added in lines 472-476.

  1. Conclusion:

Provide a concise summary of your key findings. Mention avenues for future research based on your results.

Response: The conclusion has been revised as the comments in lines 478-48

Reviewer 2 Report

Comments and Suggestions for Authors

The manuscript “The hypoglycemic and hypolipidemic effects of phenolic extract from Sargassum pallidum in vivo” by Xie Xing et al. is a complete study of the phenolic extract of Sp activities in an animal model of T2DM, its influence in the metabolic parameters affected, the composition of gut microbiota and the serum metabolomics analysis. The study results original, relevant and well presented. Some general and particular comments are listed below.

General comments

The abstract is a well description of the results obtained, however would be desirable include as introduction some previous antecedents about the Sargassum pallidum extracts activities in T2DM, glycemic and lypidemic controls. Moreover, some materials and methods also need to be presented. In general, the abstract need to be improved to reflect the complete content of the manuscript.

The introduction section includes all the relevant information about Sargassum phenolics, T2DM and molecular pathway explored in the manuscript. Just a brief paragraph expressing the aim of the study is necessary.

In the materials and methods section, the origin of the dried powder of Sargassum pallidum must be specified. About the total phenolic content determination, at least, need to be identified before be cited. In relation to the animal use, there is no mention about ethical committee approbation; the committee identification and the code of protocol approval need to be mandatory presented. When the quantitative RT-PCR assay is described, the information and trademarks about enzymes, nucleotides and other reagents is missing; this information need to be presented in the section.

The results section is complete, ordered and descriptive. Only, some suggestions are made to improve the figures presentation. In figure 1 (panels A and B) and others the letters size result to small and groups description is difficult to analyze. In addition, a clear bar pattern distinction could be included, may be, coloring the bars can help to differentiate the groups and times of treatments. Moreover, please, increase the lines thickness.

The discussion section is complete, ordered and emphasize the pertinence of the obtained results in an animal model of T2DM. There are no suggestions to change this part.

Particular comments

Line 13. Italics to Sargassum pallidum.

L20-22 and others: Genus and species must to appear in italics, i.e. Bacterioides, Lactobacillus, Enterococcus and Helycobacter.

L34 and others: include space before the cite square bracket “so on[3].”

L47 and others: “et al”, because is Latin, must to appears in italics.

L97 what model means? Untreated group?

L114 to specify the insulin ELISA kit.

L132 to specify the BCA protein assay kit.

L200 30 minutes?

L202 to mention the Fig 1F

L270-271, please control the description about the effect of SPPE on gluconeogenesis. Particularly, unify with the presented in the abstract (line 16).

Conclusion

It is a major change to do to include the ethical approbation related to the animal use. Other minor changes are necessaries and are described above.

Comments on the Quality of English Language

There is no additional comments about English quality.

Author Response

Dear reviewer,

We greatly appreciate the detailed constructive comments and suggestions from the reviewer. Those comments are very valuable and helpful for improving our paper. The manuscript has been carefully revised accordingly to the comments and suggestions are listed below.

  • The abstract is a well description of the results obtained, however would be desirable include as introduction some previous antecedents about the Sargassum pallidum extracts activities in T2DM, glycemic and lypidemic controls. Moreover, some materials and methods also need to be presented. In general, the abstract need to be improved to reflect the complete content of the manuscript.

Response: In this study, the mice model was used to investigated the hypoglycemic and hypolipidemic effect of SPPE. Therefore, the key indexes were presented in this part. Also, some methods were added.

  • The introduction section includes all the relevant information about Sargassum phenolics, T2DM and molecular pathway explored in the manuscript. Just a brief paragraph expressing the aim of the study is necessary.

Response: The aim of the study was presented in lines 79-84.

  • In the materials and methods section, the origin of the dried powder of Sargassum pallidum must be specified. About the total phenolic content determination, at least, need to be identified before be cited. In relation to the animal use, there is no mention about ethical committee approbation; the committee identification and the code of protocol approval need to be mandatory presented.

Response: The origin of the dried powder of Sargassum pallidum was supplied in lines 87-88. The total phenolic content was determined in line 191. The ethical committee approbation has been provided in lines 123-126.

  • The results section is complete, ordered and descriptive. Only, some suggestions are made to improve the figures presentation. In figure 1 (panels A and B) and others the letters size result to small and groups description is difficult to analyze. In addition, a clear bar pattern distinction could be included, may be, coloring the bars can help to differentiate the groups and times of treatments. Moreover, please, increase the lines thickness.

Response: The different patterns represent different groups, and the figures have been modified according to the comments.

  • Line 13. Italics to Sargassum pallidum.

Response: It has been revised in line 19.

  • L20-22 and others: Genus and species must to appear in italics, i.e. Bacterioides, Lactobacillus, Enterococcus and Helycobacter.

Response: It has been revised in line 29 and 30.

  • L34 and others: include space before the cite square bracket “so on[3].”

Response: They have been revised in the whole manuscript.

  • L47 and others: “et al”, because is Latin, must to appears in italics.

Response: They have been revised in the whole manuscript.

  • L97 what model means? Untreated group?

Response: Yes, it is untreated model that it has been revised.

  • L114 to specify the insulin ELISA kit.

Response: It has been revised.

  • L132 to specify the BCA protein assay kit.

Response: It has been revised.

  • L200 30 minutes?

Response: Right. It has been revised.

  • L202 to mention the Fig 1F

Response: It has been added.

  • L270-271, please control the description about the effect of SPPE on gluconeogenesis. Particularly, unify with the presented in the abstract (line 16).

Response: It has been revised in lines 272-274.

  • It is a major change to do to include the ethical approbation related to the animal use. Other minor changes are necessaries and are described above.

Response: It has been revised in line 123-126.

Reviewer 3 Report

Comments and Suggestions for Authors

Manuscript entitled The hypoglycemic and hypolipidemic effects of phenolic extract from Sargassum pallidum in vivo” is a well written paper which fortifies the understanding of the therapeutic applications Sargassum pallidum extract but also provides a platform for further explorations into its properties. I believe it has high scientific value appropriate.

Some minor sugestions are listed below:

Line 33 - “Diabetes patients with high blood level for a long time.. “ High blood level of which parameters?

The content of phenols in used extracts was expresses as gallic acid equivalents. Authors claim (from their previously published results) quercetin derivatives and 6-gingerol are the main phenolics in Sargassum pallidum extract. Is it possible to provide data concerning their content in applied extract, i.e. about doses of these specific compounds given to mice?

Clarify Acronyms and Abbreviations: The first time an acronym or abbreviation appears, it should be clearly defined. While you have done this for most, ensure it is consistent throughout.

Please check grammar and typos throughout the manuscript.

Comments on the Quality of English Language

/ 

Author Response

Dear reviewer,

We greatly appreciate the detailed constructive comments and suggestions from the reviewer. Those comments are very valuable and helpful for improving our paper. The manuscript has been carefully revised accordingly to the comments and suggestions are listed below.

  • Line 33 - “Diabetes patients with high blood level for a long time.. “ High blood level of which parameters?

Response: For the high blood level, it means the level higher than 11.1 mmol/L.

  • The content of phenols in used extracts was expresses as gallic acid equivalents. Authors claim (from their previously published results) quercetin derivatives and 6-gingerol are the main phenolics in Sargassum pallidum extract. Is it possible to provide data concerning their content in applied extract, i.e. about doses of these specific compounds given to mice?

Response: Based on previous study, the main component was determined according to the peak of LC-MS, and the amount was not determined. Therefore, the exact amount of the main components in the extracts are not sure.

  • Clarify Acronyms and Abbreviations: The first time an acronym or abbreviation appears, it should be clearly defined. While you have done this for most, ensure it is consistent throughout.

Response: All these have been checked and revised in the whole manuscript.

  • Please check grammar and typos throughout the manuscript.

Response: The grammar and typos throughout the manuscript have been checked and revised.

Reviewer 4 Report

Comments and Suggestions for Authors

It’s an exciting document that contributes to the field. However, the following comment must be considered to improve the paper.

Considering that a gram has 1000 mg- and that the SPPE contains 22.63 ± 0.21 mg GAE/g: how is the extract called “phenolic extract”? Also, it is part of the title and along the text. 22.66 is the 2.2% of the extract, which is a low content. Thus, 97% of the evaluated extracts are other compounds that are not mentioned and not discussed in the text. If the composition of the SPPE is known, it must be included and taken into account for the discussion.

The suggestion is to present the SPPE as a general extract because it’s not “rich” or “high” or an isolated of phenolic compounds. According to the phenolic content, the majority of components are unknown. Therefore, the discussion must consider it and avoid supporting all the results on phenolics. Also,  include more references giving information on the effects of other compounds of the SPPE that may be inducing the displayed results. For example, microbiota may be modulated for many other compounds. When phenolic compounds are discussed as the main responsible for the effects, references (other papers) with the doses evaluated of the isolated compounds must be included and compared to the content of this individual compound present in the doses contained in the SPPE at the evaluated doses, to really support that the mentioned compound may be the responsible of the discussed effect. In another way, the discussion is a weak supposition.

Results and discussion

The terms “control and model” are a little bit confusing- please use other terms to clearly identify the meaning of those groups.

Lines 178-179: the results on BW are not linear, so this affirmation is not really supported by results. SPPE50 displayed a higher BW decrease than SPPER250 doses; thus, please rewrite the sentence giving the correct description of the results

Figure 1- includes the meaning of OGTT

Lines 188-191: Specify that SPPE250 was the only treatment that displayed significant effects on FBG, compared to the model, which includes a value. Also, mentions that treatment (S) had a similar effect compared to Met and that none of the treatments reached the Control levels.

Lines 193-194: includes statistical significance.

Lines 196-197: considering insulin results, it must be specified that SPPE250  “was almost 

similar to metformin.”

Lines 199-200: Figure 1E doesn´t include 15 minutes, and also, the highest values were reached after 20 minutes. Furthermore, Model displayed the highest values after 80 minutes. Please correct the information.

Figure 1E- includes statistical values

Line 201- what is the meaning of “obviously”? Statistically different than?

Figure 2- the title must specify that in the effect of SPPE on fat accumulation….. (A)

Section 3.6- include p-value to identify significant differences 

Figure 3- are all “levels”? Or is the activity of the enzyme?

Line 255- avoid the use of subjective words: what´s the meaning of “considerable”?

3.8 section:  gene expression a

Lines 267- avoid the use of “remarkable” use objective words

Line 286- use T2DM, as in the text.

Figure 6 and Table 3- change the please in the text to be read first in the table

Line 294- include the F/B value

Lines 297-298: “Furthermore, the relative contents of Bacteroides and Lactobacillus were augmented by 2.52 and 0.58 times”-  include the groups

Figures 7- improve resolution (they are blurry)

Figure 7G- is it not possible to identify each group as the text mentions (Model and SPPE250 groups)- or have the same results?

Lines 376-378: include the doses of the phenolic compounds and, if available, the evaluated individual compounds.- How is it compared to the compounds present at the doses given in this study?

Lines 401-404- what are the proposed compounds that are inducing the changes?

Lines 428-429 and 432-434 stated that “SPPE was -abundant- in phenolics like 6-gingerol and quercetin derivatives.” therefore, the restoration of these mRNA 432 expressions of glycolipid metabolism-related in T2DM mice may be due to the synergetic 433 combinations effect of phenolics in Sargassum pallidum”. However, phenolics are not the primary compounds in the evaluated extract.

As part of the discussion, it is necessary to identify what of the generated metabolites come from the SPPE components. The discussion mentions fatty acids but is not related to the extract composition.

Line 491- How can the used animal model support it (treat and prevent)? “for the utilization of SPPE to treat and prevent T2DM”. It’s not aligned with the conclusion of the abstract (management of T2DM- line 24.

In general- avoid the use of the words “obviously” and “remarkable” in the text. Correct finger mistakes; English must be reviewed.

References

It detected inappropriate citations in the text. I have not the  time to review the correct use (citation) of each reference, but here is an example:

Lines 34 to 36- Here, it is stated that: “Simultaneously, a large number of diabetes cases showed high blood lipid profiles, which was the key factor 35 to result in cardiovascular diseases”. However, the cited paper aimed to characterize and investigate the polysaccharide fractions of mulberry fruit and their antioxidant and hypoglycemic activities, concluding the antioxidant and hypoglycemic activities of Mulberry fruit polysaccharides, indicating the need for their further exploration as potential antidiabetic agents. Thus, the text of lines 34 and 36 is not supported by the included cite.

This is a very worrying error in the text. I don't know how many times this is happening in this document. 

Comments on the Quality of English Language

Moderate editing of English language required

Author Response

Dear reviewer,

We greatly appreciate the detailed constructive comments and suggestions from the reviewer. Those comments are very valuable and helpful for improving our paper. The manuscript has been carefully revised accordingly to the comments and suggestions are listed below.

  • It’s an exciting document that contributes to the field. However, the following comment must be considered to improve the paper. Considering that a gram has 1000 mg- and that the SPPE contains 22.63 ± 0.21 mg GAE/g: how is the extract called “phenolic extract”? Also, it is part of the title and along the text. 22.66 is the 2.2% of the extract, which is a low content. Thus, 97% of the evaluated extracts are other compounds that are not mentioned and not discussed in the text. If the composition of the SPPE is known, it must be included and taken into account for the discussion. The suggestion is to present the SPPE as a general extract because it’s not “rich” or “high” or an isolated of phenolic compounds.

Response: We greatly agree with the comments. Therefore, we called Sargassum pallidum extract which has been revised in the title and content.

  • The terms “control and model” are a little bit confusing- please use other terms to clearly identify the meaning of those groups.

Response: This has been revised that with “untreated model” in line 113.

  • Lines 178-179: the results on BW are not linear, so this affirmation is not really supported by results. SPPE50 displayed a higher BW decrease than SPPER250 doses; thus, please rewrite the sentence giving the correct description of the results

Response: It has been revised in lines 2032-204

  • Figure 1- includes the meaning of OGTT

Response: This has been added in Figure 1.

  • Lines 188-191: Specify that SPPE250 was the only treatment that displayed significant effects on FBG, compared to the model, which includes a value. Also, mentions that treatment (S) had a similar effect compared to Met and that none of the treatments reached the Control levels.

Response: It has been revised in lines 213-214.

  • Lines 193-194: includes statistical significance.

Response: It has been revised in lines 217 and 219.

  • Lines 196-197: considering insulin results, it must be specified that SPPE250 “was almost similar to metformin.”

Response: It has been revised in lines 221 and 222.

  • Lines 199-200: Figure 1E doesn´t include 15 minutes, and also, the highest values were reached after 20 minutes. Furthermore, Model displayed the highest values after 80 minutes. Please correct the information.

Response: It has been revised in line 224.

  • Figure 1E- includes statistical values

Response: This has been added in line 228.

  • Line 201- what is the meaning of “obviously”? Statistically different than?

Response: The “obviously” has been changed to “significantly”. Statistically different than 4 weeks before itself.

  • Figure 2- the title must specify that in the effect of SPPE on fat accumulation…(A)

Response: It has been added in Fig 2.

  • Section 3.6- include p-value to identify significant differences

Response: This has been provided in lines 256-257.

  • Figure 3- are all “levels”? Or is the activity of the enzyme?

Response: It refers to the SOD, CAT and GSH enzymes activities.

  • Line 255- avoid the use of subjective words: what´s the meaning of “considerable”?

Response: It has been changed to “high”.

  • Lines 267- avoid the use of “remarkable” use objective words

Response: It has been deleted.

  • Line 286- use T2DM, as in the text.

Response: It has been revised in line 296.

  • Line 294- include the F/B value

Response: The value has been added in line 304.

  • Lines 297-298: “Furthermore, the relative contents of Bacteroides and Lactobacillus were augmented by 2.52 and 0.58 times”- include the groups

Response: It has been added in lines 309-310.

  • Figures 7- improve resolution (they are blurry)

Response: It has been improved.

  • Figure 7G- is it not possible to identify each group as the text mentions (Model and SPPE250 groups)- or have the same results?

Response: The model and SPPE250 group were identified for SPPE250 group had the most positive effects on diabetic mice.

  • Lines 376-378: include the doses of the phenolic compounds and, if available, the evaluated individual compounds.- How is it compared to the compounds present at the doses given in this study?

Response: In the previous our study, the phenolic compounds that quercetin-3-O-glucuronide, 6-gingerol and poricoic acid A are the main components had strong inhibitory effects on the digestive enzymes. Therefore, they also contribute to the effects in this study.

  • Lines 401-404- what are the proposed compounds that are inducing the changes?

Response: According to the previous research and literatures, the proposed compounds may be phenolic compounds.

  • Lines 428-429 and 432-434 stated that “SPPE was -abundant- in phenolics like 6-gingerol and quercetin derivatives.” therefore, the restoration of these mRNA expressions of glycolipid metabolism-related in T2DM mice may be due to the synergetic combinations effect of phenolics in Sargassum pallidum”. However, phenolics are not the primary compounds in the evaluated extract.

Response: We greatly agree with your comment. Therefore, the presentation has been changed that with bioactive compounds instead of phenolic compounds in lines 428-429.

  • Line 491- How can the used animal model support it (treat and prevent)? “for the utilization of SPPE to treat and prevent T2DM”. It’s not aligned with the conclusion of the abstract (management of T2DM- line 24.

Response: It has been revised.

  • In general- avoid the use of the words “obviously” and “remarkable” in the text.

Response: The word like “obviously” and “remarkable” have been revised in the whole manuscript.

  • It detected inappropriate citations in the text., here is an example: Lines 34 to 36- Here, it is stated that: “Simultaneously, a large number of diabetes cases showed high blood lipid profiles, which was the key factor 35 to result in cardiovascular diseases”. However, the cited paper aimed to characterize and investigate the polysaccharide fractions of mulberry fruit and their antioxidant and hypoglycemic activities, concluding the antioxidant and hypoglycemic activities of Mulberry fruit polysaccharides, indicating the need for their further exploration as potential antidiabetic agents.

Response: This citation has been revised and the whole manuscript has been checked.

Round 2

Reviewer 2 Report

Comments and Suggestions for Authors

All the comments and suggestions made previously were attended and corrected. The manuscript is now suitable for publication in Foods.

Author Response

Thanks

Reviewer 4 Report

Comments and Suggestions for Authors

After checking the R1 document, there are still important points to attend to. Several comments were not addressed. The most worrying thing is the incorrect use of references; It is indicated that it was attended to, but the text does not have the correction.

Comments can be read as “Rev”

(1) It’s an exciting document that contributes to the field. However, the following comment must be considered to improve the paper. Considering that a gram has 1000 mg- and that the SPPE contains 22.63 ± 0.21 mg GAE/g: how is the extract called “phenolic extract”? Also, it is part of the title and along the text. 22.66 is the 2.2% of the extract, which is a low content. Thus, 97% of the evaluated extracts are other compounds that are not mentioned and not discussed in the text. If the composition of the SPPE is known, it must be included and taken into account for the discussion. The suggestion is to present the SPPE as a general extract because it’s not “rich” or “high” or an isolated of phenolic compounds.
Response: We greatly agree with the comments. Therefore, we called Sargassum pallidum extract which has been revised in the title and content.

Rev: isn’t correctly modified along the text- for example, page 2-line 66; page 5, line 188. What is the composition of the extract?

(2) Theterms“controlandmodel”arealittlebitconfusing-pleaseuseothertermsto clearly identify the meaning of those groups.
Response: This has been revised that with “untreated model” in line 113.

Rev: line 113 does not have this information. Also, now, only in this line the model is referred to as “untreated” (line 183), but the text still uses “control” and “model”. Furthermore- lines 93-95 do not name the groups as in the figures.

(3) Lines 178-179: the results on BW are not linear, so this affirmation is not really supported by results. SPPE50 displayed a higher BW decrease than SPPER250 doses; thus, please rewrite the sentence giving the correct description of the results Response: It has been revised in lines 2032-204

Rev: The referred lines do not have this information

(4) Figure 1- includes the meaning of OGTT
Response: This has been added in Figure 1.
(5) Lines 188-191: Specify that SPPE250 was the only treatment that displayed significant effects on FBG, compared to the model, which includes a value. Also, mentions that treatment (S) had a similar effect compared to Met and that none of the treatments reached the Control levels.
Response: It has been revised in lines 213-214.
(6) Lines 193-194: includes statistical significance.
Response: It has been revised in lines 217 and 219.
(7) Lines196-197:consideringinsulinresults,itmustbespecifiedthatSPPE250“was almost similar to metformin.” 

Response: It has been revised in lines 221 and 222.

(8) Lines 199-200: Figure 1E doesn ́t include 15 minutes, and also, the highest values were reached after 20 minutes. Furthermore, Model displayed the highest values after 80 minutes. Please correct the information.
Response: It has been revised in line 224.

Rev: the referred line doesn´t have this information. I can´t find the information

(9) Figure 1E- includes statistical values
Response: This has been added in line 228.

(10)Line 201- what is the meaning of “obviously”? Statistically different than? Response: The “obviously” has been changed to “significantly”. Statistically different than 4 weeks before itself.

Rev: The text still displaying “obviously” (example: line 413)

(11)Figure 2- the title must specify that in the effect of SPPE on fat accumulation...(A) Response: It has been added in Fig 2.

(12)Section 3.6- include p-value to identify significant differences
Response: This has been provided in lines 256-257.
(13)Figure 3- are all “levels”? Or is the activity of the enzyme?
Response: It refers to the SOD, CAT and GSH enzymes activities.

Rev: The figure indicates that are levels- correct

(14) Line 255- avoid the use of subjective words: what ́s the meaning of “considerable”? Response: It has been changed to “high”.

Rev: The word “Considerable” is still used (line 225)

(15)Lines 267- avoid the use of “remarkable” use objective words
Response: It has been deleted.

Rev: The word “remarkable” is still used (line 391)

(16)Line 286- use T2DM, as in the text.
Response: It has been revised in line 296.

(17)Line 294- include the F/B value
Response: The value has been added in line 304.
(18)Lines 297-298: “Furthermore, the relative contents of Bacteroides and Lactobacillus were augmented by 2.52 and 0.58 times”- include the groups 
Response: It has been added in lines 309-310.
(19)Figures 7- improve resolution (they are blurry)
Response: It has been improved.
(20)Figure 7G- is it not possible to identify each group as the text mentions (Model and SPPE250 groups)- or have the same results?
Response: The model and SPPE250 group were identified for SPPE250 group had the most positive effects on diabetic mice.
(21)Lines 376-378: include the doses of the phenolic compounds and, if available, the evaluated individual compounds.- How is it compared to the compounds present at the doses given in this study?
Response: In the previous our study, the phenolic compounds that quercetin-3-O- glucuronide, 6-gingerol and poricoic acid A are the main components had strong inhibitory effects on the digestive enzymes. Therefore, they also contribute to the effects in this study. 

Rev: This information must be part of the discussion. How is it compared to the compounds present at the doses given in this study? amount?

(22)Lines 401-404- what are the proposed compounds that are inducing the changes? Response: According to the previous research and literatures, the proposed compounds may be phenolic compounds.

(23)Lines 428-429 and 432-434 stated that “SPPE was -abundant- in phenolics like 6- gingerol and quercetin derivatives.” therefore, the restoration of these mRNA expressions of glycolipid metabolism-related in T2DM mice may be due to the synergetic combinations effect of phenolics in Sargassum pallidum”. However, phenolics are not the primary compounds in the evaluated extract. 

Response: We greatly agree with your comment. Therefore, the presentation has been changed that with bioactive compounds instead of phenolic compounds in lines 428- 429.

(24)Line 491- How can the used animal model support it (treat and prevent)? “for the utilization of SPPE to treat and prevent T2DM”. It’s not aligned with the conclusion of the abstract (management of T2DM- line 24. 

Response: It has been revised.
(25)In general- avoid the use of the words “obviously” and “remarkable” in the text. 
Response: The word like “obviously” and “remarkable” have been revised in the whole manuscript.
(26)It detected inappropriate citations in the text., here is an example: Lines 34 to 36- Here, it is stated that: “Simultaneously, a large number of diabetes cases showed high blood lipid profiles, which was the key factor 35 to result in cardiovascular diseases”. However, the cited paper aimed to characterize and investigate the polysaccharide fractions of mulberry fruit and their antioxidant and hypoglycemic activities, concluding the antioxidant and hypoglycemic activities of Mulberry fruit polysaccharides, indicating the need for their further exploration as potential antidiabetic agents.
Response: This citation has been revised and the whole manuscript has been checked. 

Rev: The text contains the same mistake- the reference was not corrected, thus, there is a high possibility that the whole document was not checked.

Author Response

Dear reviewer,

We greatly appreciate the detailed constructive comments and suggestions from the reviewer. Those comments are very valuable and helpful for improving our paper. The manuscript has been carefully revised accordingly to the comments and suggestions are listed below.

  • Isn’t correctly modified along the text- for example, page 2-line 66; page 5, line 188. What is the composition of the extract?

Response: We are sorry there are some missing. And all these have been revised. The extract should contain phenolic compounds, polysaccharides, protein, alkaloids, microelement, etc.

  • line 113 does not have this information. Also, now, only in this line the model is referred to as “untreated” (line 183), but the text still uses “control” and “model”. Furthermore- lines 93-95 do not name the groups as in the figures.

Response: We all seriously discussed this and have added the explanation for the model in lines 114-115 which will avoid the confusion. And we also found related reports using model.   

  • The referred lines do not have this information.

Response: For revision in the front part, it was displayed in lines 204-205.

  • The referred line doesn´t have this information. I can´t find the information.

Response: For revision in the front part, it was displayed in lines 224-226.

  • The text still displaying “obviously” (example: line 413).

Response: It has been revised.

  • The word“Considerable” is still used (line 225).

Response: It has been revised.

  • The word “remarkable” is still used (line 391).

Response: It has been revised.

  • This information must be part of the discussion. How is it compared to the compounds present at the doses given in this study? amount?

Response: It is compared based on the composition and content that quercetin-3-O- glucuronide, 6-gingerol and poricoic acid A has been identified in the Sargassum pallidum extracts with the highest content. The information has been added in the lines 368-371.

  • The text contains the same mistake- the reference was not corrected.

Response: We really appreciate your comments. This research is about hypoglycemic effects of Sargassum pallidum extracts. Some researches are about polysaccharide, but also focused on the hypoglycemic effect. The information about diabetes were cited. Therefore, there may be some confusing from the title. Than you for your suggestions.